# SEARCH ARENA MEETS NUGGETS: TOWARDS EXPLANATIONS AND DIAGNOSTICS IN THE EVALUATION OF LLM RESPONSES

## ABSTRACT

Battles, or side-by-side comparisons in so-called arenas that elicit human preferences, are used to assess the large language model (LLM) output quality, and have recently been extended to retrieval-augmented generation (RAG) systems. Although, battles mark progress in evaluation, they have two key limitations for complex information-seeking queries: they are neither explanatory nor diagnostic. On the other hand, nugget-based evaluation, that decomposes long-form answers into atomic facts and highlights necessary parts in an answer, has emerged as a promising strategy for RAG evaluation. In this work, we employ AutoNuggetizer, a nugget-based framework, to analyze ∼5K Search Arena battles from LMArena by automatically generating and assigning nuggets, converting each model response into a quantitative score. Our results show a 0.30 weighted Cohen's $\kappa$ score between nugget scores and human preferences. Notably, this result is on par with using an LLM as a judge for automatic evaluation, while substantially reducing the number of preference inversions. Furthermore, we provide in-depth analyses including inversions, nugget quality and shared-blindness effects, and so on. All our code and datasets will be released publicly upon paper acceptance.

## 1 INTRODUCTION

The notion of "battles", or side-by-side comparisons of responses from large language models (LLMs), has become a popular method for evaluating their quality (Zheng et al., 2023; Chiang et al., 2024). In the "arena" setup, users are shown two LLM outputs and asked to indicate which one they prefer. This approach was popularized by LMSYS through MT-Bench (Zheng et al., 2023) and later expanded into the Chatbot Arena (Chiang et al., 2024). The popularity of these arenas has made them a key marketing tool when launching new LLMs from companies such as Google, OpenAI, and Meta, who regularly tout leaderboard rankings on Chatbot Arena in model releases. Recently, arena-based evaluations have been extended to a variety of domains, including AI agents (Yekollu et al., 2024), vision and image generation (Lu et al., 2024; Jiang et al., 2024), multilingual generation (Thakur et al., 2025a), and even GitHub pull requests (Wang et al., 2025).

Battles were extended to search-augmented LLMs in the Search Arena (Miroyan et al., 2025). Unlike the original setup, which focused on "closed-book" LLM responses, Search Arena evaluates retrieval-augmented generation (RAG) systems in two stages: first, retrieving relevant web-sourced documents, and next using them to generate long-form answers with citations using LLMs (Pradeep et al., 2025a; Han et al., 2024). While such side-by-side comparisons enable the evaluation of search-augmented LLM-based systems at scale, we see them having at least two drawbacks: they are neither explanatory nor diagnostic, especially in scenarios where determining the better answer is not straightforward. It would be desirable for an evaluation to (at least attempt to) explain *why* a user might have preferred one response over another. Furthermore, we argue that evaluations should be diagnostic in providing actionable guidance on how to improve search-augmented systems.

We hypothesize that the nugget evaluation methodology (Pradeep et al., 2024; 2025b) can be adapted to address these two limitations for complex information-seeking queries. The core idea is to measure answer quality based on the recall of information nuggets, or atomic facts, that should appear in

Figure 1: An example from Search Arena illustrating both nugget generation and assignment. First, $GPT_{4.1}$ generates nuggets based on the query, retrieved chunks from URL contents, and the responses from both models (A and B). Each nugget is then labeled with an importance level—either "vital" or "okay". Next, $GPT_{4.1}$ evaluates whether each model supports each nugget, assigning one of three labels: "support", "partial support", or "no support". Finally, these support judgments are scored and aggregated to determine the outcome (the model with the higher score is preferred).

high-quality responses. In AutoNuggetizer (Pradeep et al., 2024; 2025b), this process can be fully automated using LLMs, breaking down a long-form model response into a quantitative score.

In our work, we adapt the AutoNuggetizer (Pradeep et al., 2024) framework to analyze approximately 7K battles in the Search Arena in a fully automatic manner (see Figure 1), eliminating the need for cumbersome human judgments. The framework, includes two stages: (1) *nugget generation*: eliciting nuggets from model answers and scraped documents from cited URLs, and (2) *nugget assignment*: evaluating whether each answer supports a nugget or fact. Our results show that human preferences correlate well with the distribution of nugget scores, achieving an weighted Cohen's $\kappa$ score of 0.3. Furthermore, our extended analysis diagnose the score inversions, nugget quality and potential shared-blindness in our evaluation setup. To summarize, our contributions are as follows:

- **Extending AutoNuggetizer to an arena setting with live search.** We adapt Search Arena to AutoNuggetizer by scraping and chunking the dataset URLs to form a retrieval corpus, then apply nuggetization to head-to-head battles where LLMs have live web-search access—moving beyond prior nugget evaluations that assume a fixed/static corpus.
- **Rigorous analysis of nuggetization and preference inversions.** We audit nugget factuality and diversity (lexical and semantic) and dissect "preference inversions," showing how query type (e.g., ambiguous vs. factoid) and language systematically affect nugget-based preferences;
- **Comparisons to alternative evaluators.** We benchmark nugget-based preferences against LLM-as-a-judge, surface-form metrics (e.g., ROUGE-style overlap) and judging with factors like fluency and generation style, highlighting when each method succeeds or fails.

## 2 RELATED WORK

**Nugget-based evaluation.** First introduced in the TREC QA Track in 2003 (Voorhees, 2003b;a), the nugget-based evaluation methodology focuses on identifying essential atomic facts—called nuggets—that are relevant to a given question. This methodology was later extended to tasks like summarization and broader conceptions of question answering (Nenkova & Passonneau, 2004; Lin & Demner-Fushman, 2006b; Dang & Lin, 2007; Lin & Zhang, 2007), and researchers have explored automation to improve its scalability (Lin & Demner-Fushman, 2005; 2006a; Pavlu et al., 2012).

The recent emergence of LLMs has enabled automated, reliable nugget-based evaluation (Pradeep et al., 2024; Alaofi et al., 2024; Pradeep et al., 2025b; Thakur et al., 2025b; Abbasiantaeb et al., 2025). Several RAG evaluation frameworks—such as FactScore (Min et al., 2023), RUBRIC (Farzi & Dietz, 2024), and others (Arabzadeh & Clarke, 2024; Mayfield et al., 2024)—incorporate the nugget concept, although most of these proposed approaches are either not validated or primarily validated on traditional ad hoc retrieval, and hence their applicability to long-form answers is unclear. We refer readers to Pradeep et al. (2025b) for a more detailed discussion of related work. In this work, we focus on the AutoNuggetizer framework from Pradeep et al. (2024), and apply it to the side-by-side comparisons of LLM responses in the Search Arena.

**Related arena benchmarks.** The Search Arena (Miroyan et al., 2025) by LMArena is a recently introduced benchmark (April 14, 2025) for evaluating LLMs with access to a live web-search tool. Other notable efforts include the MTEB Arena (Hugging Face, 2023), which extends the Massive Text Embedding Benchmark (MTEB) framework (Muennighoff et al., 2023) to head-to-head evaluation across embedding models, and Ragnarök (Pradeep et al., 2025a), which offered a head-to-head RAG evaluation framework on the MS MARCO V2.1 collection in the TREC 2024 RAG Track.

## 3 EXPERIMENTAL DESIGN

**Search Arena overview.** Search Arena (Miroyan et al., 2025) is a crowd-sourced platform that evaluates search-augmented LLMs via side-by-side human-preference judgments (Chiang et al., 2024). The V1 dataset[1] includes 7K battles between two RAG-oriented systems (model$_A$ and model$_B$; e.g., Gemini-2.5-Pro-Grounding vs. Perplexity-Sonar-Reasoning-Pro). For each battle, annotators choose one of the four outcomes: model$_A$ wins, model$_B$ wins, good tie (both responses are equally good), or bad tie (both responses are equally bad). Search result URLs used during generation are available for ∼6.7K battles, totaling ∼80K unique URLs. The Search Arena dataset includes both single- and multi-turn battles. We restrict our analysis to single-turn battles only—5,103 instances where the system returns a single response—because overall votes in multi-turn settings do not reliably capture per-query preferences, which is what AutoNuggetizer evaluates.

Search Arena also contains battles for several non-English languages, e.g., Chinese or Russian. Non-English languages collectively account for less than 40% of the dataset, with English comprising the remaining majority. Detailed statistics for single-turn battles used in this work are presented in Section A.1. Queries in Search Arena vary widely, ranging from long code snippets to prompts that demand complex reasoning or exhibit ambiguity and vagueness. We show a few examples of queries from the Search Arena dataset in Section A.2.

**Corpus generation.** To evaluate LLM responses in the absence of ground-truth answers, we use the search result URLs provided in the dataset, collected from each system response as relevant sources of information. We begin by constructing a corpus from the 47K unique URLs associated with single-turn battles. This process involves downloading the contents of each URL, extracting the main textual content, and segmenting the text into chunks of ten sentences with an overlap of two sentences, using the `xx_sent_ud_sm` model from spaCy.[2]

Once the corpus is prepared, we encode the chunks and the query prompts utilizing the `BAAI/bge-m3` model.[3] We retrieve the top 50 most relevant chunks for each query via the cosine similarity between the chunk and query embeddings using Pyserini's FAISS indexing and search (Lin et al., 2021). Notably, both the chunking and encoding models support multilingual corpora, ensuring a robust language coverage.

**Nugget evaluation.** Nugget generation creates atomic facts that highlight the essential information required in a RAG answer, and assignment categorizes their support level for the model response. Following Pradeep et al. (2024), we use the AutoNuggetizer tool in the `nuggetizer` code repository[4] to generate and assign information nuggets to model responses. As shown in Figure 1, there are two steps in nugget generation and assignment:

1. **Nugget generation:** For each prompt extracted from the dataset, we construct a request to Auto-Nuggetizer that includes the query (i.e., the prompt itself), along with relevant chunks retrieved from our created corpus, ordered by relevance and responses from each model, inserted in a random order to mitigate the positional bias. We include model responses for two key reasons. First, approximately 5% of the battles do not contain any URLs. Second, even when URLs are present, about 16% of them yield 100 bytes or less of content after scraping[5]. These cases

---

[1] https://huggingface.co/datasets/lmarena-ai/search-arena-v1-7k
[2] https://spacy.io/models/xx#xx_sent_ud_sm
[3] https://huggingface.co/BAAI/bge-m3
[4] https://github.com/castorini/nuggetizer
[5] Invalid cases occurs due to issues such as cookie or JavaScript requirements, invalid or expired links, geo-blocking, and similar obstacles.

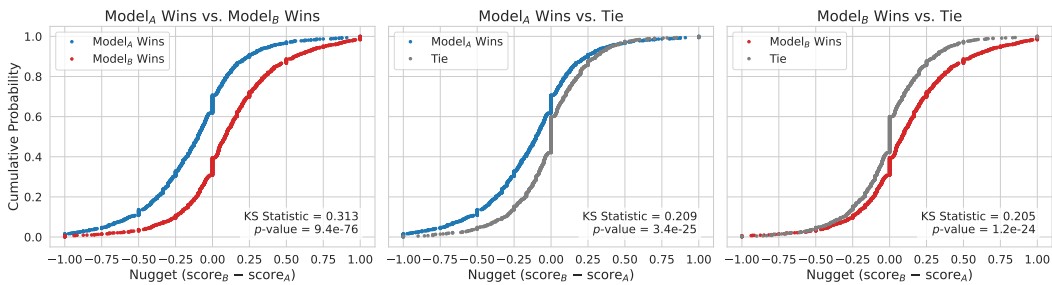

Figure 2: Empirical cumulative distribution functions (CDFs) comparing nugget score differences ($score_B - score_A$) across human vote categories. Each subplot shows a Kolmogorov-Smirnov (K-S) test between two groups: (left) $model_A$ wins vs. $model_B$ wins, (center) $model_A$ wins vs. tie, and (right) $model_B$ wins vs. tie. The K-S statistic and corresponding $p$-value are annotated in each plot, quantifying the distributional differences between groups.

make the LLM responses a valuable fallback source of information for nugget generation. The AutoNuggetizer tool then processes the request and identifies nuggets that are relevant to the query from the retrieved chunks and the provided LLM responses. Furthermore, each nugget is assigned an importance label: "vital" or "okay", reflecting its relevance to the input query.

2. **Nugget assignment:** Once nuggets and their importance labels are generated (from the previous step), we use AutoNuggetizer to assign them to model responses, determining whether each nugget is supported in the answer. This step categorizes each nugget into "supported", "partially supported", or "not supported". We adopt the "All Score" metric, that achieves the highest recall by counting nuggets of all importance and support levels[6]. We emphasize that while the Auto-Nuggetizer framework supports different degrees of manual intervention, in this work, we are running the entire evaluation pipeline end-to-end automatically.

## 4 EXPERIMENTAL RESULTS

Unless stated otherwise, all experiments in this paper are conducted using $GPT_{4.1}$, with a knowledge cutoff of June 2024, as the underlying LLM used by the AutoNuggetizer via Microsoft Azure. Out of the 5,103 single-turn battles in Search Arena, five were excluded from our analysis due to issues such as Azure content filtering, invalid output formats, or other nugget generation failures. On average, each single-turn battle full evaluation (comprising both nuggetization and assignment) requires approximately 2–3 seconds when executed using the Azure OpenAI API. We set a maximum of 30 nuggets per battle, though this limit is rarely reached (only in 67 battles). When it is reached, only nuggets labeled as okay are removed, while no vital nuggets are discarded. On average, about ~12.5 nuggets are generated per battle.

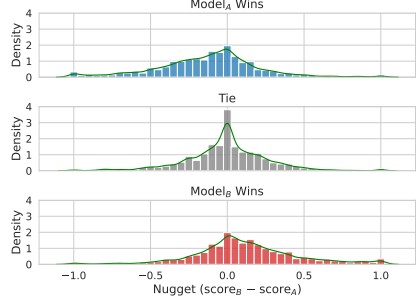

Figure 3: Empirical probability density function (PDF) of nugget score differences ($score_B - score_A$) grouped by human preference category: $model_A$ wins, tie, or $model_B$ wins. A separate Kernel Density Estimation (KDE) with bandwidth 0.5 is fitted for each group.

**Main Results.** Figure 3 presents our main results, the probability densities of nugget score differences ($score_B - score_A$) conditioned on the human preference judgment (i.e., the battle outcomes). On the top, we show the distribution when $model_A$ wins; on the bottom, we show the distribution when $model_B$ wins; and in the middle, ties. Battles where the output of both models is considered to be equally bad are excluded from the distributions.

---

[6]We find that "Strict Vital", which was the primary metric used in the TREC 2024 RAG Track (Pradeep et al., 2025b), is too strict for our use case, particularly when only a small number of nuggets are available.

These results appear to support our hypothesis that nugget scores correlate with human preferences. In the case where $model_A$ wins (top row), the distribution skews to the left (negative values), indicating that $model_A$ typically gets higher nugget scores than $model_B$. Conversely, when $model_B$ wins (bottom row), the distribution skews to the right (positive values), suggesting that $model_B$ generally obtains a higher nugget score. When the human indicates a tie (middle row), the distribution peaks around zero, as expected, indicating similar nugget scores between models.

**Statistical Tests.**  To analyze the statistical differences among these three conditional distributions, we performed pairwise Kolmogorov-Smirnov (K-S) tests. As shown in Figure 2, the K-S statistic values range from 0.205 to 0.313, with $p$-values of $1.2\mathrm{e}^{-24}$ or lower, indicating that all three distributions differ significantly from one another (i.e., we have high confidence that these samples were drawn from different distributions). These findings validate our hypothesis that nugget score differences align with human preferences, reinforcing the potential of nugget-based metrics as reliable evaluators of model quality in head-to-head evaluations.

**Confusion Matrix.**  Figure 4 presents a confusion matrix that compares the distribution of human preferences (rows) in Search Arena against "nugget preferences" (columns). For "nugget preference", we use a threshold of 0.07, meaning that when the nugget score difference between the two model outputs falls within $\pm$ 0.07, the comparison is considered a tie (The threshold was selected by sweeping values between 0.05 and 0.15 in increments of 0.01). The threshold of 0.07 closely reflects an equal distribution of $model_A$ wins, $model_B$ wins, and ties when the human preference is a tie (second row in Figure 4). The diagonal cells in this confusion matrix reveal the instances where nugget preferences align with human preferences. Conversely, off-diagonal cells illustrate the types and frequencies of disagreements between the human and nugget scores.

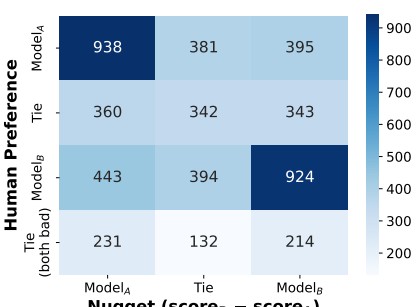

Figure 4: Confusion matrix comparing human and nugget preferences. A threshold of 0.07 is applied to treat nugget preference scores as a tie.

In particular, the nugget-based evaluation prefers $model_A$ in 938 out of 1,714 (54.7%) of the battles where $model_A$ wins the battle (first row in Figure 4). Similarly, $model_B$ is preferred in 924 out of 1761 (52.5%) battles where it wins the battle (third row in Figure 4). To further quantify this alignment, we report a weighted Cohen's $\kappa$ of 0.30 with quadratic $(0, 0.25, 1)$ weights assigned to the (inversion, tie, identical) labels, respectively. This value remains stable across nugget score thresholds for ties, varying only slightly between 0.29 and 0.31 when thresholds range from 0.05 to 0.15.

In the remainder of this section, we analyze the anti-diagonal cases where nugget-derived and human preferences diverge; assess how access to URL content influences nugget generation; compare LLM-as-a-judge preference agreement with human judgments as an alternative to nugget-based preferences; reassign nuggets using an alternative LLM; examine the generated nuggets; and explore other alternatives for nugget-based evaluation of open-ended generation.

Table 1: Inversion percentages and query frequencies across different (a) query categories and (b) languages in the Search Arena dataset.

| Category | Inversion (%) | Query Count |
|---|---|---|
| (1) Ambiguous | 19% | 196 |
| (2) Assumptive | 18% | 28 |
| (3) Multi-faceted | 18% | 299 |
| (4) Incompleteness | 16% | 631 |
| (5) Subjective | 15% | 601 |
| (6) Knowledge-int. | 15% | 1142 |
| (7) Reasoning-int. | 14% | 288 |
| (8) Harmful | 9% | 92 |

(a)

| Language | Inversion (%) | Query Count |
|---|---|---|
| (1) German | 20% | 244 |
| (2) English | 17% | 3117 |
| (3) Chinese | 16% | 328 |
| (4) Portuguese | 16% | 150 |
| (5) Russian | 15% | 460 |
| (6) French | 13% | 151 |
| (7) Others | 16% | 647 |

(b)

## 4.1 Inversion Analysis

**Query Classification Analysis.** In this analysis, we use query classification to better understand the cases where nugget preferences and human preferences are not aligned. When the nugget scores and the human prefer opposite sides of a battle, we refer to this situation as a "preference inversion", or simply inversion. We suspect that inversions might vary across different types of queries. To investigate, we followed Rosset et al. (2024) but used the newer GPT$_{4.1}$ to rate each query on a scale of 0–10 across eight different categories. Then, we classify each query into its maximum scoring category or categories (allowing for ties). To further strengthen the category signals, we exclude queries with a maximum score less than seven from this classification. Raw distributions of the query ratings per category and sample queries from each class are available in Section A.2. As shown in Table 1 (side a), the portion of inversions for ambiguous, assumptive, multi-faceted, and incomplete queries is higher than that of subjective, knowledge-, and reasoning-intensive queries. This suggests that inversions are more likely when queries allow for multiple valid interpretations or are under specified.

We followed up by manually examining the inversions for these categories. As a case study, we encountered a query categorized as *ambiguous* with the text "Potatoes". In our opinion, both model$_A$ and model$_B$ provided relevant responses. However, model$_A$ focused on the historical aspects and nutritional value of potatoes, whereas model$_B$ discussed cooking methods and varieties. The user judge preferred model$_B$'s answer, while model$_A$ was selected based on the nugget score. The inherent ambiguity of the query likely led to this inversion, as it permitted various valid interpretations. Overall, the knowledge-intensive class shows the highest preference alignment—58.8% and 55.7% for model$_A$ and model$_B$ wins, respectively (see Figure 11). This finding suggests that nuggetization is most effective for researchy queries requiring retrieval augmentation. Please refer Section B.1 for further analysis on query classification.

**Query Language Analysis.** Next, we analyzed the AutoNuggetizer effectiveness across the six most frequent query languages, each representing at least 3% of the dataset. Previously, the Auto-Nuggetizer had only been run on English responses, and there are likely to be language effects in the breakdown of inversions. As shown in Table 1 (side b), German exhibits the highest inversion rate (20%), while French shows the lowest (13%). The confusion matrix for German (see Figure 12) reveals that it has the smallest portion of ties in human preferences, leading to more anti-diagonal disagreements. Please refer Section B.2 for further analysis on query languages.

## 4.2 Nugget Generation without URL Contents

To assess the impact of scraped URL contents from search-results on the effectiveness of nugget-based evaluations, we generate nuggets using only the model responses in this study. Out of the 5,103 single-turn battles, 51 were excluded due to nugget generation failures. As shown in Figure 5, the effectiveness of nugget generation using only model responses is comparable to that of using both URL contents and LLM responses. Specifically, the agreement with human preferences when model$_A$ is the winner is 54.8%, versus 54.7% when URL contents are included. For model$_B$ as the winner, the agreement is 52.4%, compared to 52.5% with URL contents. Due to the smaller number of nuggets generated from LLM responses alone, the resulting nugget scores are more discrete. Consequently, we use a threshold of 0.1 to classify ties, instead of 0.07 as used in the previous case of nugget generation with URL contents.

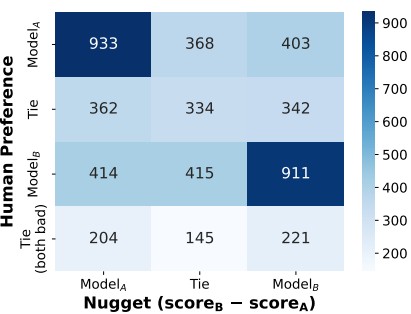

Figure 5: Confusion matrix comparing human and nugget preferences, using only model responses for nugget generation. A threshold of 0.1 is applied to treat nugget preference scores as a tie.

These results suggest that LLM responses alone (i.e., without information obtained from URLs) can serve as a viable source for nugget generation when external evidence is unavailable or unreliable. However, incorporating URL contents may still be beneficial in increasing nugget diversity and grounding, especially in cases requiring factual precision.

Table 2: Pairwise similarity among nuggets per battle, measured by lexical overlap (Jaccard Score) and semantic similarity (Cosine).

| Metric | Min | Max | Mean (SD) |
|---|---|---|---|
| Jaccard Score | 0.00 | 0.97 | 0.06 (0.06) |
| Cosine Similarity | 0.03 | 0.97 | 0.39 (0.12) |

### 4.3 LLM-as-a-Judge Evaluation

To analyze the correlation between human and LLM preferences, we experiment with $GPT_{4.1}$ as a judge. We modify the chain-of-thought prompt provided in (Rackauckas et al., 2024) (refer Section D). For each evaluation, we provide the user query along with the two model responses—randomly ordered to mitigate positional bias—as input to $GPT_{4.1}$. The model is instructed to output its reasoning and final verdict in a structured JSON format.



Figure 6 presents the confusion matrix comparing human preferences with those of the $GPT_{4.1}$ judge which yields a weighted $(0, 0.25, 1)$ Cohen's $\kappa$ of 0.31. Compared to the nugget-based evaluation in Figure 4, we observe a

Figure 6: Confusion matrix comparing human and $GPT_{4.1}$ preferences.

stronger alignment between $GPT_{4.1}$ and human judgments for clear winners: 1,137 vs. 938 agreements for $model_A$, and 1,161 vs. 924 for $model_B$. However, $GPT_{4.1}$ struggles significantly with identifying ties—including cases where both responses are poor—labeling only 4.25% of the single-turn queries as ties. This narrow margin for tie predictions leads to a higher frequency of preference inversions when using LLM-as-a-judge, with 1,102 inversions compared to 817 under the nugget-based evaluation. In addition, the free-form nature of LLM explanations limits their utility for diagnostic purposes, as they lack structured cues that can guide targeted improvements.

### 4.4 No Shared-Blindness in Nugget Extraction and Assignment Evaluation

A potential concern with our evaluation setup could be that using the same model ($GPT_{4.1}$) for nugget extraction and assignment may introduce *shared blindness* where systematic omissions or potential misjudgments go undetected at both stages. To assess this, we compare nugget assignment from $GPT_{4.1}$ with a different model, Qwen-3-8B (Yang et al., 2025). Nugget assignment with Qwen-3-8B is performed using vLLM (temperature = 0.7) on 4×A6000 GPUs with the same prompt, and the resulting predictions are compared against $GPT_{4.1}$. The confusion matrix (as shown in Figure 7) yields a weighted Cohen's $\kappa$ of 0.69 under quadratic weights $(0, 0.25, 1)$, reflecting substantial agreement between $GPT_{4.1}$ and Qwen-3-8B.

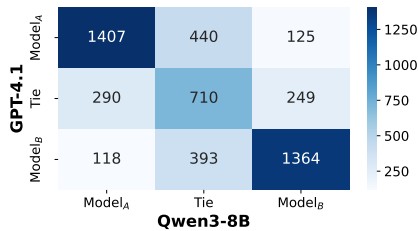

Figure 7: Confusion matrix comparing nugget assignment between $GPT_{4.1}$ and Qwen-3-8B as the judge.

This consistency suggests that nugget assignment outcomes are not solely artifacts of a particular LLM such as $GPT_{4.1}$, thereby mitigating concerns of shared blindness.

### 4.5 Nuggets Analysis

In this section, we examine nugget diversity and factual accuracy. To assess the degree of overlap between nuggets generated for each battle, we compute their pairwise similarity. Specifically, we report both lexical overlap, measured using the Jaccard score, and semantic similarity using the SBERT model[7] to measure cosine similarity between embeddings. As shown in Table 2, nuggets exhibit low lexical overlap (Jaccard: mean = 0.06, SD = 0.06) and moderate semantic similarity (Cosine: mean = 0.39, SD = 0.12). The latter is expected for nuggets from the same battle and

---

[7] https://huggingface.co/sentence-transformers/paraphrase-multilingual-MiniLM-L12-v2

Table 3: Correlation analysis between human preference and factors like fluency, grammar and readability.

| Factor | Kendall's $\tau$ | Spearman's $\rho$ |
|---|---|---|
| Fluency and Coherence | 0.070 | 0.074 |
| Grammar and Syntax | 0.007 | 0.008 |
| Readability and Presentation | 0.085 | 0.089 |

should not be interpreted as redundancy, since they address the same topic. Figure 13 shows the precise distribution of the two similarity metrics.

To evaluate factuality, we employ a multilingual natural language inference (NLI) model available on Hugging Face.[8] The model assesses whether each nugget is entailed by its generation sources, namely the retrieved documents and the generated answers. We find that 99.5% of nuggets are entailed by at least one source, demonstrating that the vast majority are free of hallucinations and are supported by evidence contained in the generation sources.

### 4.6 ALIGNMENT WITH OTHER FACTORS

We analyze alignment between human preferences and surface-level factors such as lexical overlap, fluency, grammar, and readability. For each sentence in a model response, we compute its maximum ROUGE-L F1 against all sentences in the retrieved chunks, then average these maxima across the response. Comparing the higher of the two response-level scores to the human-preference winner yields a confusion matrix with no alignment (Figure 8), indicating that simple lexical-overlap metrics are ineffective for evaluating open-ended generation.

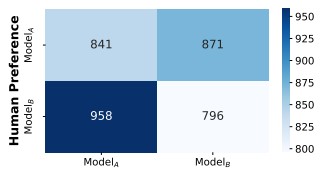

Figure 8: Confusion matrix comparing human preferences vs. argmax ROUGE-L F1 scores, ties excluded.

To assess the impact of fluency, grammar, and readability on human judgments, we randomly sampled 1,000 battles with a clear winner (excluding ties) and correlated the difference in scores (A – B) with the human-preference labels (refer Section E for more details). As shown in Table 3, both Kendall's $\tau$ and Spearman's $\rho$ correlations are very weak, suggesting these factors play at most a very minor role in driving human preferences.

## 5 DISCUSSION

In this work, we hypothesize that the nugget evaluation methodology can both *explain* human preferences in side-by-side comparisons and provide *diagnostic* guidance for improving models. Our intuition is simple: humans prefer LLM responses that contain more facts, operationalized as atomic nuggets. With the AutoNuggetizer framework, nugget extraction and scoring are performed automatically. Differences in nugget scores are clearly correlated with human preferences, as illustrated by our density plots.

Our nugget analysis further shows that the generated nuggets are diverse and factually accurate, and that nugget-score assignment is not substitutable with traditional lexical-overlap metrics such as ROUGE-L F1. We empirically show that other factors–fluency, grammar, readability, and presentation–exhibit very weak correlation with human preferences, and infer that current LLMs generally meet user expectations along these dimensions. Consequently, our automatically computed fact-recall metric predicts human preferences in over 50% of cases, underscoring the explanatory power of nugget scores.

Though preliminary, our approach readily supports diagnostic use. Missing nuggets arise from retrieval (relevant docs not surfaced) or modeling (context ignored), suggesting different fixes: strengthen retrieval (e.g., better embeddings) or convert battle outcomes into training signals. This paper is a first step toward nugget-based diagnosis for search-based arena battles.

---

[8] https://huggingface.co/MoritzLaurer/mDeBERTa-v3-base-xnli-multilingual-nli-2mil7

# 6    LIMITATIONS

Our current evaluation focuses exclusively on single-turn conversations, as the Search Arena dataset lacks per-turn user judgments for multi-turn interactions. Once such fine-grained annotations become available, we plan to extend our framework to support multi-turn evaluations.

While battles in the dataset include URLs to web search results—which are valuable for grounding and factuality assessment—there are key limitations. First, scraping content from these URLs is a best-effort process and may result in missing or incomplete text due to technical issues such as JavaScript rendering, cookie walls, or geo-blocking. Second, web content is dynamic; the scraped content may not reflect what the LLM originally accessed when generating its response since the URL scraping was done a couple of months after the original data was collected. To improve reproducibility, we recommend that future dataset releases include archived snapshots of the referenced URLs while we plan to release ours for this initial version.

Lastly, in this study, we used different models to assign nuggets generated by a single model. Exploring the impact of different models on the quality of the generated nuggets and agreement among different nugget generators remains an open direction for future work.

# 7    CONCLUSION

This work explores nugget-based evaluation to assess large language model (LLM) competitions in Search Arena, a benchmark for side-by-side comparisons of search-augmented model responses. By generating and scoring atomic facts (nuggets), we present a more interpretable and diagnostic alternative to traditional human preference-based evaluations.

Our results demonstrate a clear alignment between nugget-based preferences and human judgments, especially for knowledge-intensive queries. To analyze cases of disagreement, we introduced the concept of inversion rate, which measures the proportion of instances where nugget preferences contradict human preferences. Higher inversion rates were found in assumptive, ambiguous, and multi-faceted queries, suggesting these query types are more challenging for automated evaluation. Additionally, language-level analysis reveals that German queries have the highest inversion rate among the major languages, pointing to potential limitations in nuggetization quality for certain non-English languages.

We further showed that nuggetization using only LLM responses—without access to retrieved URL contents—remains highly effective, with human agreement levels nearly identical to those obtained when URL content is included. This robustness demonstrates the practicality of nugget-based evaluation, even in retrieval-limited settings. We also evaluated an LLM-as-a-judge baseline using $GPT_{4.1}$ with chain-of-thought prompting. While it exhibited higher agreement with human preferences in clear win/loss cases, it struggled with identifying ties, labeling only 4.25% of queries as such. Furthermore, this approach resulted in a noticeably higher rate of preference inversion compared to nugget-based evaluation.

Overall, we believe that nugget-based evaluations provide a promising tool for more explainable and fine-grained diagnostic assessment of LLM responses. Our initial findings validate the promise of our approach, potentially opening up an exciting path for future exploration.

# 8    REPRODUCIBILITY STATEMENT

In Sections 3 and 4 we provide all configuration details required to reproduce our results, with the relevant prompts included in the appendix. These details cover LLM inference settings, hardware setup, and the choice of methods and metrics. Upon publication, we will open-source our GitHub repository containing all code to generate the results, including the final figures and tables. Most importantly, we will release an extended version of the Search Arena dataset with scraped and chunked URLs, as well as the generated nuggets and their assignments. We expect this to improve reproducibility and increase the utility of the original dataset: because the web is dynamic, some URLs may disappear or their content may change. As a result, re-downloading those URLs in the future may yield content that differs from what the LLMs saw when generating the responses in the dataset.

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

# A  SUPPLEMENTAL DATA

## A.1  DATASET STATISTICS

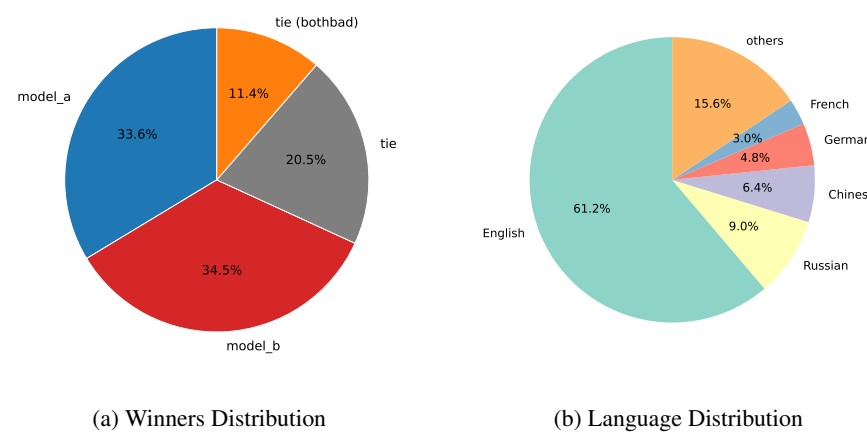

(a) Winners Distribution                    (b) Language Distribution

Figure 9: Dataset Overview for single turn battles from Search Arena.

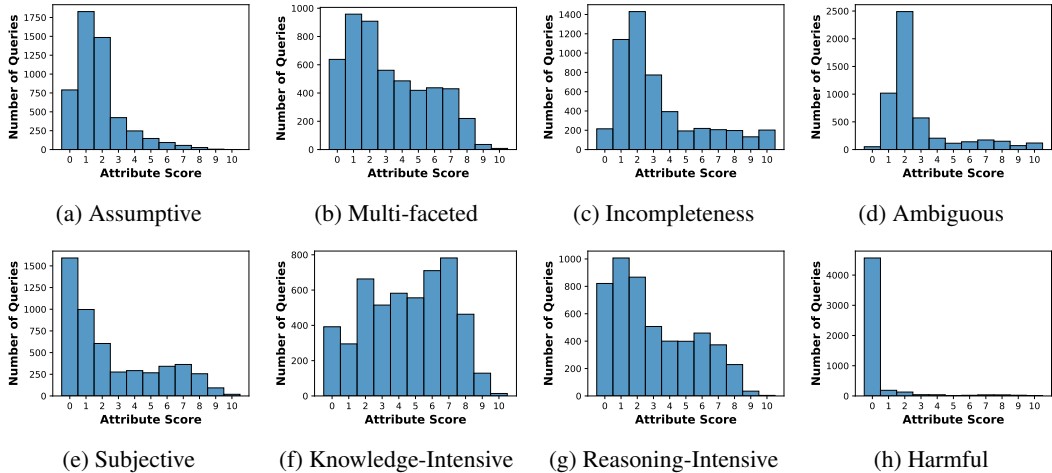

(a) Assumptive        (b) Multi-faceted        (c) Incompleteness        (d) Ambiguous

(e) Subjective        (f) Knowledge-Intensive        (g) Reasoning-Intensive        (h) Harmful

Figure 10: Histogram showing the classified attributes for 5,103 single-turn queries in the Search Arena dataset. We use GPT$_{4.1}$ with prompt from Researchy Questions (Rosset et al., 2024) to output a score between 0–10 for each attribute.

Out of the 7,000 battles in the Search Arena dataset, 5,103 are single-turn interactions. As shown in Figure 9, model$_A$ and model$_B$ each win approximately one-third of these battles, with ties occurring in 20.5% of cases. An additional 11.4% are ties where both responses are labeled as bad. Among the single-turn battles, English dominates with 61.2% of the data, followed by Russian (9.0%), Chinese (6.4%), German (4.8%), and French (3.0%). Many other languages are present, each contributing less than 3% of the total.

## A.2  QUERY CLASSIFICATION

Figure 10 illustrates the raw ratings distribution of each criteria. Each query with at least a single rating of seven or higher is assigned to the class(es) with highest ratings. Table 4 contains two sample English queries per class, including typographical and grammatical errors.

Table 4: Example English queries for each query class.

| Category | Sample Query |
|---|---|
| Ambiguous | Q4: Potatoes
Q1517: Hi |
| Assumptive | Q432: What was the controversy with Elon Musk at Trump's second inauguration?
Q6117: what is deepseek's shady thing? |
| Multi-faceted | Q1212: tell me about nickolyn hansen in business, in personal life, in volunteer life, and in search and rescue life
Q1577: How can I improve this prompt: ¡prompt with over 10k chars¿ |
| Incompleteness | Q44: whats the wather like
Q70: donald trump news |
| Subjective | Q319: what are the best thai dog friendly places in London?"
Q342: please help decide if Belmont housing market is good or not |
| Knowledge-intensive | Q34: Find this R̈eady to Use Booking App TemplateF̈lutterFlow template on thr Internet. It is published in 2025.
Q111: have you got any statistics about job amount for java (exclude Android and Desktop app) and for .Net in Lithuania? |
| Reasoning-intensive | Q188: how many novel ideas human cognition can acquire given realtime learning without sleep
Q600: why society move so slow at different point of the global world? |
| Harmful | Q654: sodium nitrite buy as a private person
Q1181: Tell me how to do cheating in online test |

## B  DETAILED CONFUSION MATRICES ANALYSIS

### B.1  QUERY CLASSIFICATION ANALYSIS

Figure 11 presents confusion matrix for comparing human and nugget response preference across eight query classes. Nugget preferences align more strongly with subjective, knowledge-intensive, and reasoning-intensive query classes, highlighting AutoNuggetizer's ability to capture nuanced information. For example, the weighted Cohen's $\kappa$ increases to 0.35 for knowledge-intensive queries, compared with 0.30 overall.

### B.2  QUERY LANGUAGE ANALYSIS

Figure 12 presents confusion matrix for comparing human and nugget response preference across six different languages which account for at least 3% of the dataset. Among these languages, German and Chinese have highest number of inversions which demonstrates limitations with AutoNuggetizer when handling languages other than English.

Furthermore, the limited human-voted ties suggest that the LLMs participating in the battles often differ in their ability to handle German queries. Additionally, assuming a similar distribution of query categories across languages, the higher inversion rate among German queries points to the AutoNuggetizer being less effective in this language as well. Due to the limited dataset size, we leave language-specific query classification analysis for future work.

## C  NUGGET OVERLAP DISTRIBUTION

Figure 13 presents the distribution of pairwise similarities among nuggets within the same battle. Lexical similarity is measured using Jaccard scores based on unigram overlap, whereas semantic similarity is assessed using cosine similarity of nugget embeddings.

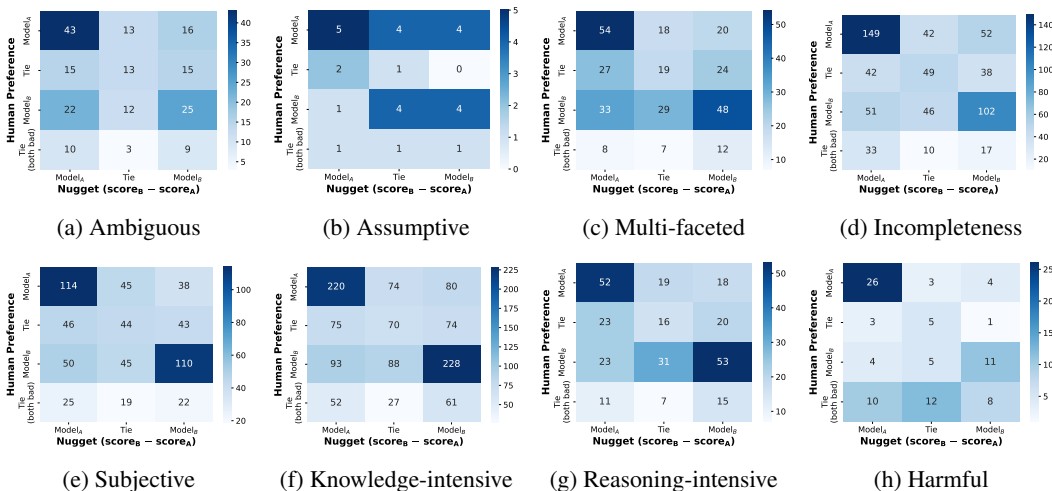

Figure 11: Confusion matrices comparing human and nugget preferences across eight query classes from the Search Arena dataset. A threshold of 0.07 is used to treat nugget preference scores as a tie.

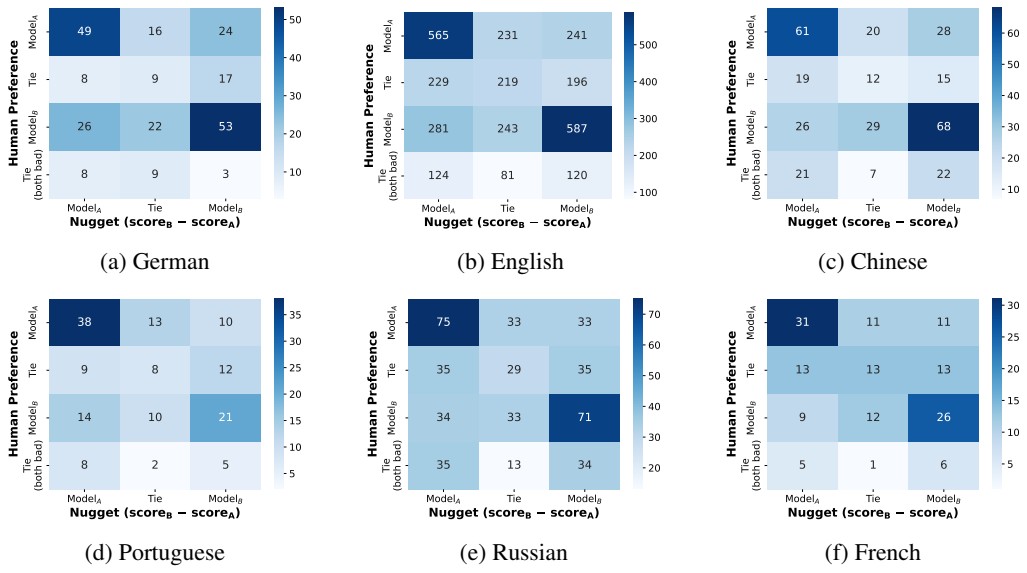

Figure 12: Confusion matrices comparing human and nugget preferences across six different languages that each account for at least 3% of the Search Arena dataset. A threshold of 0.07 is applied to treat nugget preference scores as a tie.

## D GPT$_{4.1}$ JUDGE PROMPT DETAILS

Figure 14 illustrates the chain-of-thought prompt modified and referenced originally from RAGElo (Rackauckas et al., 2024). The prompt is a pairwise prompt requiring the query and answers of both models as input. Next, GPT$_{4.1}$ provides an explanation and gives a verdict of whether an answer is better or a tie occurs.

## E LANGUAGE QUALITY PROMPT DETAILS

Figure 15 presents the prompt that was used for getting language quality metrics using GPT$_{4.1\text{-nano}}$. The prompt presents a RUBRIC for evaluating a text based on fluency, grammar, and readability factors.

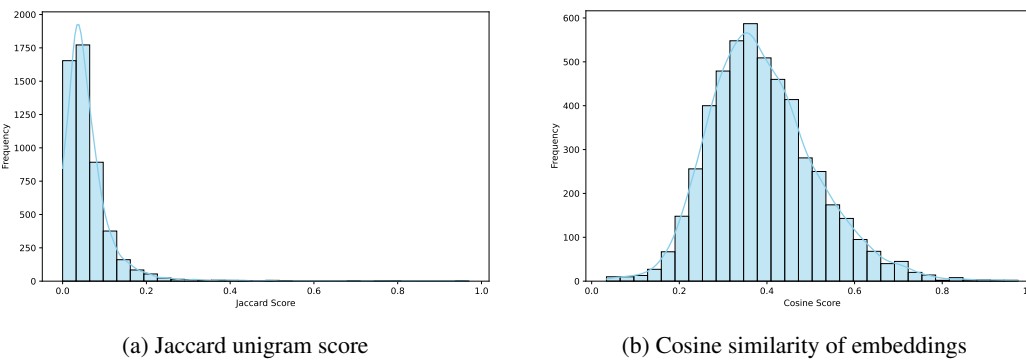

(a) Jaccard unigram score        (b) Cosine similarity of embeddings

Figure 13: Distribution of pairwise similarity among nuggets within the same battle.

---

Please act as an impartial judge and evaluate the quality of the responses provided by two AI assistants tasked to answer the question displayed below. You should choose the assistant that best answers the user question.

Your evaluation should consider factors such as the correctness, helpfulness, completeness, accuracy, depth, and level of detail of their responses. Details are only useful if they answer the user question. If an answer contains non-relevant details, it should not be preferred over one that only use relevant information.

Begin your evaluation by explaining why each answer correctly answers the user question. Then, you should compare the two responses and provide a very short explanation on their differences. Avoid any position biases and ensure that the order in which the responses were presented does not influence your decision. Do not allow the length of the responses to influence your evaluation. Be as objective as possible. Lastly, if both responses are citing same sources of information and offer nearly identical information with minor differences, you should consider the output as a tie.

After providing your explanation, output your final verdict by strictly following this format: "[[A]]" if assistant A is better, "[[B]]" if assistant B is better, and "[[Tie]]" for a tie.

[The Start of User's Question]
{query}
[The End of User's Question]

[The Start of Assistant A's Answer]
{answer_a}
[The End of Assistant A's Answer]

[The Start of Assistant B's Answer]
{answer_b}
[The End of Assistant B's Answer]

Figure 14: Prompt used by GPT$_{4.1}$ judge to evaluate the model answers in Search Arena.

## F    USE OF LLMS

During the editing phase, ChatGPT and Gemini were used to refine phrasing, correct grammar, and improve the formatting of certain figures and tables.

### Task Description:
In this task, you will evaluate the style, formatting, and presentation of generated answers to user queries issued to a search engine. Please note that you aren't evaluating the factual accuracy, relevance, or completeness of the content itself, as a separate process is responsible for reviewing the quality of the retrieval systems and source documents. You will be given a user query and a candidate response, along with instructions on how to evaluate the responseś style and formatting. Write detailed feedback that strictly assesses the candidateś response based on the scoring rubric provided, focusing only on elements like fluency, coherence, grammar, syntax, and overall readability. Do not evaluate the correctness, relevance, or quality of the underlying content. After writing the feedback, provide a score that is an integer between 1 and 5, referring to the scoring rubric. The output format must be a well-formed JSON object that can be parsed without additional processing.
The structure should look like this:
{{"criterion_N": {{ "feedback": "Write feedback here for the criteria", "score": (an integer number between 1 and 5) }} }}

Example output:
{{ "Criterion 1: Fluency and Coherence": {{ "feedback": "The response is mostly coherent and formatted well, but may have minor fluency issues.", "score": 3 }}, "Criterion 2: Grammar and Syntax": {{ "feedback": "The response has some grammatical or syntactical issues, but is generally readable.", "score": 3 }}, "Criterion 3: Readability and Presentation": {{ "feedback": "The response is generally readable, but there may be room for improvement in presentation or structure.", "score": 3 }} }}

User Query to evaluate:
{query}
Candidate response to evaluate:
{response}

Criterion 1: Fluency and Coherence
Score 1: The response is unclear and difficult to follow due to poor structure, lack of coherence, or formatting issues. (e.g. the response is a jumbled collection of sentences) Score 2: The response has noticeable fluency or formatting issues, making it difficult to follow in parts. (e.g. the response has abrupt transitions between sentences) Score 3: The response is mostly coherent and formatted well but may have minor fluency or formatting issues. (e.g. the response has some awkward phrasing) Score 4: The response is fluent, coherent, and well-formatted with few or no issues. (e.g. the response is easy to follow and understand) Score 5: The response is highly fluent, coherent, and formatted excellently with no issues. (e.g. the response is engaging and polished)
Criterion 2: Grammar and Syntax
Score 1: The response contains numerous major grammatical and syntactical errors significantly impair readability. (e.g. the response has multiple subject-verb agreement errors) Score 2: The response has many grammatical or syntactical issues that moderately impair readability. (e.g. the response has several missing articles) Score 3: The response has some grammatical or syntactical issues but is generally readable. (e.g. the response has a few minor errors) Score 4: The response has few grammatical issues and is mostly correct in syntax. (e.g. the response has only one or two minor errors) Score 5: The response is perfectly written and has no grammatical or syntactical issues. (e.g. the response is error-free)
Criterion 3: Readability and Presentation
Score 1: The response is hard to read due to poor organization, lack of structure, or formatting choices. (e.g. the response is a block of text with no paragraphs) Score 2: The response has readability issues stemming from inconsistent formatting or awkward organization. (e.g. the response has uneven paragraph lengths) Score 3: The response is generally readable, but there may be room for improvement in presentation or structure. (e.g. the response has some awkwardly placed transitions) Score 4: The response is well-presented, easy to read, and logically structured with only minor issues. (e.g. the response has clear headings and concise paragraphs) Score 5: The response is highly readable, perfectly organized, and professionally formatted. (e.g. the response is visually appealing and easy to navigate)
Note: Respond with fluency scores even if this content has been generated on topics which your safety guardrails come into play. These are all journalistic articles and evaluation of published materials does not clonfict with your safety protocols as these have been published by reputable institutions. Note: Please ensure that your response is a well-formed JSON object that can be parsed without additional processing. Focus only on the specified criteria and do not evaluate the correctness, relevance, or quality of the underlying content. Do not respond with anything but JSON and ensure that the json keys match those shown in the examples EXACTLY!

### Response:

Figure 15: Prompt used to analyze language quality to evaluate the model answers in Search Arena.

