# OpenReview forum: "Search Arena Meets Nuggets: Towards Explanations and Diagnostics in the Evaluation of LLM Responses"
_ICLR.cc/2026/Conference — ICLR 2026 Conference Withdrawn Submission_

### Official Review · Reviewer_tuaU · 2025-10-28

**Soundness:** 3
**Presentation:** 3
**Contribution:** 2
**Rating:** 4
**Confidence:** 4

**Summary:**

1. This paper proposes applying the AutoNuggetizer framework to the Search Arena dataset, aiming to provide explanatory and diagnostic evaluations for search-augmented LLMs.

2. The paper conducts extensive experiments to examine whether automated nugget-based evaluation can serve as a substitute for human preference judgments, including correlation and comparative analyses.

3. The experimental results show that nugget-based evaluation is a reliable automatic alternative and can also generalize well to settings without retrieval.

**Strengths:**

1. This paper tests across ~5k single-turn battles in Search Arena with multiple analyses—statistical tests, inversion breakdown, language effects, and model ablations—showing rigorous experimental demonstration.

2. The paper considers potential biases introduced by using LLM-as-Judge, designs dedicated consistency experiments, and observes a strong alignment.

3. Clear experimental details and dataset augmentation (scraped and chunked URLs) enhance the paper’s credibility and community utility.

**Weaknesses:**

1. This work is entirely built upon AutoNuggetizer, which limits its novelty and contribution. Moreover, to the best of my knowledge, AutoNuggetizer adopts an iterative nugget extraction procedure that maintains a nugget list of size k and continuously updates it while traversing the chunks. This may lead to later-extracted nuggets overwriting earlier, useful ones, potentially causing substantial coverage loss.

2. The corpus used in this paper is extracted from the URLs cited by the search-augmented LLMs in Search Arena. However, since the evaluated LLMs may not access all relevant webpages, this could result in missing nuggets and introduce bias into the evaluation. In particular, the harder the query, the less complete the information retrieved by the evaluated LLMs, and the higher the evaluation noise.

3. The search-augmented LLMs studied in this work are limited to single-turn settings, ignoring the increasingly popular multi-turn agentic search paradigm.

**Questions:**

Please refer to Weaknesses.

---

### Official Review · Reviewer_ngeS · 2025-11-01

**Soundness:** 2
**Presentation:** 2
**Contribution:** 2
**Rating:** 2
**Confidence:** 3

**Summary:**

This submission proposes integrating nugget-based evaluation into the Search Arena framework for retrieval-augmented LLM evaluation. The authors adapt the AutoNuggetizer system to automatically generate and assign atomic factual units (“nuggets”) from ~7K Search Arena battles, achieving a weighted Cohen’s κ of 0.30 with human preferences — comparable to GPT-4.1-as-judge baselines. The study also provides analyses of inversion cases, multilingual effects, and "shared blindness" issues, arguing that nugget-based metrics offer both explanatory and diagnostic advantages.

**Strengths:**

This work draws attention to ponder the explanability of pairwise comparison when conducting human evaluation of LLM responses. It proposed using AutoNuggetizer for LLM + RAG result RAG evaluation and find human-AutoNuggetizer achieved 0.3 cohen kappa correlation.

**Weaknesses:**

1. There is a crucial weakness of this submission: this submission lacks a key independent technical contribution (algorithm, framework, dataset or software) to be accepted by main conference. Though correctly cited, AutoNuggetizer contributes mostly to the analysis and conclusions of this submission.  It might be suitable for a workshop.
2. Confusion Related Work (line 106): the authors had better draw an efficient abstract/conclusion of AutoNuggetizer's related work rather than redirecting readers to works outside the paper context. Also, since AutoNuggetizer is applied, the authors had better formally introduce it as a preliminary.

3. The authors had better list the equations about the Nugget Evaluation score calculation before the experiment section.

4. As an evaluation work, this work lacks abundent baselines for comparisons. As I comprehend, this work  addresses the superority of AutoNuggetizer when evaluating LLM responses to long-form QAs when grounded by retrieval information. The authors should reproduce more baslines/frameworks other than direct LLM-as-a-judge.

   Also, when reporting results, there should be a main result table, gathering the human-method  pearson/Kendall/spearman correlation or cohen kappa. Currently, such score is either omitted or seperated in Section 4 (line 246-251).

5. The authors shall provide more in-depth analysis into the accessibilty of AutoNuggetizer in LLM response evaluation. Specifically, specify the research gap claim 'they are neither explanatory nor diagnostic (line 17)'. Currently the conclusions of the submission is weak.

6. 0.3 cohen kappa is relatively low for this method. Consider analyze those cases that disagreed by human judgments.

**Questions:**

See weakness

---

### Official Review · Reviewer_F8Bx · 2025-11-03

**Soundness:** 3
**Presentation:** 3
**Contribution:** 1
**Rating:** 2
**Confidence:** 4

**Summary:**

The paper extends the battle-style evaluation framework from LMSys and Chatbot Arena to the RAG setting. The authors leverage nuggets (atomic facts extracted from scraped and chunked dataset URLs) to build a retrieval corpus that supports head-to-head LLM battles with live web search access. Their goal is to move beyond prior nugget evaluations limited to static corpora. They report a weighted Cohen’s κ of 0.30 between nugget scores and human preferences, and provide analyses of nugget factuality and diversity. The paper also highlights preference inversions, focusing on how factors such as query type (e.g., ambiguous vs. factoid) and language systematically influence preferences for nuggets.

**Strengths:**

The fundamental strong point of this current work was the Preference Inversion analysis. The discussion around which categories of prompts have the chance to make the system contradict human preference, along with the exploration of the reason, was a great idea in the paper, which should have been explored even further with stronger manual error case analysis.

The authors provide a strong argument about their system capabilities by showcasing the right set of statistical analyses, especially where their approach seems to match the human distribution.

**Weaknesses:**

1. The paper shows a notable lack of novelty. The approach is heavily dependent on the AutoNuggetizer, with the primary contribution being its application to the RAG evaluation setting. This feels more like a straightforward transfer of an existing algorithm to a new context rather than a substantive methodological advancement.
2. I am curious to know why only GPT was used as the LLM as a Judge model. It would have been great to see at least some analysis with other state-of-the-art models
3. I believe that an LLM-as-a-judge approach, with more effective prompting, could have outperformed the AutoNuggetizer method. For instance, by prompting the LLM to explicitly extract “nuggets” following the authors’ own definition, it could generate structured factual units and use them to compute the same metrics employed in the paper. Moreover, since the AutoNuggetizer automation by the authors themselves relies on LLM outputs, it is unclear why a well-prompted LLM baseline was not considered. Given that modern LLMs can handle substantially longer contexts, such a baseline would have provided a more convincing and informative comparison.
4. In Table 1, the distribution of task categories helps explain why the AutoNuggetizer approach appears to perform well. The knowledge-instance tasks are highly fact-oriented, which naturally favors a nugget-based method that leverages atomic factual elements from the context. However, this raises concerns about the generalizability of the results since the evaluation seems skewed toward knowledge-intensive tasks that inherently benefit this approach. *Most of the statistical analysis needs to be taken with a pinch of salt because of this overwhelming skewness in the dataset*
5. Additionally, the paper should report the performance of GPT (as a Judge) on the same task distributions to provide a more balanced and comparative perspective.

**Questions:**

Line 137-139: "This process involves downloading the contents of each URL, extracting the main textual content, and segmenting the text into chunks of ten sentences with an overlap of two sentences, using the xx sent ud sm model from spaCy.": Could the authors justify the reason for using this specific config

Line 351: I disagree with the authors’ claim that the free-form nature of LLM explanations limits their diagnostic utility. In practice, LLMs can reliably produce structured outputs, such as JSON-formatted explanations that provide clear, interpretable cues for targeted analysis and model improvement. (Suggestion)


Overall, after reading the AutoNuggetizer work, I found the contribution level of this paper to be relatively limited, even in terms of statistical analysis. While the authors attempt to demonstrate a new use case for the existing algorithm, the paper lacks substantive methodological or analytical advancements. For a venue like ICLR, one would expect more significant innovation or deeper empirical insights beyond incremental adaptations and minor extensions of prior work.

---

### Official Review · Reviewer_zCLp · 2025-11-03

**Soundness:** 2
**Presentation:** 3
**Contribution:** 2
**Rating:** 4
**Confidence:** 5

**Summary:**

This paper introduces the nugget-based evaluation framework into the Search Arena benchmark for comparing retrieval-augmented LLMs. It automatically extracts atomic factual units, i.e. nuggets, from each question, the retrieved evidence chunks, and the two model responses, then determines which nuggets are supported by each response to produce an aggregated nugget score. The differences in nugget scores correlate with human preferences on about 5K single-turn Search Arena examples. The authors further analyze this correlation across query types, languages, and the presence or absence of URL evidence, and compare against a GPT-4.1-as-judge baseline. They argue that nugget-based evaluation provides a more interpretable and diagnostic alternative to standard preference voting.

**Strengths:**

1. The paper extends nugget-based evaluation to the Search Arena setting, enabling more interpretable analyses of model comparison results.
2. The authors conduct extensive analyses and robustness checks, including with/without URL evidence, across languages and query categories, and against a GPT-4.1 judging baseline.
3. The planned release of the annotated dataset will be a useful contribution for researchers interested in automatic evaluation of retrieval-augmented LLMs.

**Weaknesses:**

1. The measured agreement (Cohen’s K = 0.3) indicates only fair alignment, not strong correlation. While statistically significant, this suggests that nugget scores explain only a small portion of human preferences, limiting the usefulness of the proposed method. Moreover, the paper does not address how this correlation could meaningfully reduce human annotation effort in practice, especially when they are not aligned for an example.
2. Although the paper claims that nuggets make evaluation explanatory, it reports only aggregate results such as distributions and score differences. There is no qualitative or case-based analysis showing how specific nuggets explain why one model wins or loses. Without such examples, the “explainability” claim remains theoretical.
3. The paper reports inversion statistics across query categories, but omits the key factor of answer correctness. It would be important to analyze how alignment between human and nugget judgments changes for cases where one response is correct and the other is wrong, versus when both responses are partially correct. This would clarify where nuggetization truly tracks human evaluation.
4. The “without-URL” experiment produces nearly identical results to the “with-URL” case (agreement 54.8% vs. 54.7%), implying that retrieved evidence barely influences nugget generation. This raises concerns about whether the framework meaningfully evaluates retrieval-grounded generation which is the core goal in RAG evaluation.

**Questions:**

1. How does correlation vary by answer correctness? For example, does nugget alignment strengthen when one answer is factually correct and the other is not?
2. What does the near-identical result in the “without-URL” setting imply? How does this evaluate RAG systems.
3. How can nuggets concretely assist human analysis?

---

### Note · Authors · 2025-11-28

I have read and agree with the venue's withdrawal policy on behalf of myself and my co-authors.